# Screening for Metal-Chelating Activity in Potato Protein Hydrolysates Using Surface Plasmon Resonance and Peptidomics

**DOI:** 10.3390/antiox13030346

**Published:** 2024-03-13

**Authors:** Mads Bjørlie, Julie Christina Hartmann, Line Hyrup Rasmussen, Betül Yesiltas, Ann-Dorit Moltke Sørensen, Simon Gregersen Echers, Charlotte Jacobsen

**Affiliations:** 1National Food Institute, Technical University of Denmark, 2800 Kgs. Lyngby, Denmark; mabjo@food.dtu.dk (M.B.); juliechartmann@gmail.com (J.C.H.); linerasmussen2@gmail.com (L.H.R.); betye@food.dtu.dk (B.Y.); adms@food.dtu.dk (A.-D.M.S.); 2Department of Chemistry and Bioscience, Aalborg University, 9220 Aalborg, Denmark; sgr@bio.aau.dk

**Keywords:** antioxidant peptides, sequential hydrolysis, mass spectrometry, bioinformatics, Ni-NTA binding, emulsions, lipid oxidation

## Abstract

Metal-catalyzed lipid oxidation is a major factor in food waste, as it reduces shelf life. Addressing this issue, our study investigates the potential of hydrolysates derived from potato protein, a by-product of potato starch production, as metal-chelating antioxidants. Through sequential enzymatic hydrolysis using alcalase or trypsin combined with Flavourzyme, we produced various hydrolysates, which were then fractionated using ultrafiltration. Using a combination of peptidomics and bioinformatics, we predicted the presence of metal-chelating and free radical-scavenging peptides across all hydrolysate fractions, with a trend indicating a higher content of antioxidant peptides in lower molecular weight fractions. To validate these predictions, we utilized surface plasmon resonance (SPR) and a 9-day emulsion storage experiment. While SPR demonstrated potential in identifying antioxidant activity, it faced challenges in differentiating between hydrolysate fractions due to significant standard errors. In the storage experiment, all hydrolysates showed lipid oxidation inhibition, though not as effectively as ethylenediaminetetraacetic acid (EDTA). Remarkably, one fraction (AF13) was not significantly different (*p* < 0.05) from EDTA in suppressing hexanal formation. These results highlight SPR and peptidomics/bioinformatics as promising yet limited methods for antioxidant screening. Importantly, this study reveals the potential of potato protein hydrolysates as antioxidants in food products, warranting further research.

## 1. Introduction

Along with microbial spoilage, lipid oxidation is one of main causes of food waste due to reduced shelf life by decreasing the nutritional value and sensory quality of foods [1]. One strategy to mitigate lipid oxidation is the use of antioxidants, and research into identifying natural and sustainable food antioxidants is gaining increased attention [2]. Lipid oxidation is catalyzed by transition metal ions like Fe^3+^/Fe^2+^, which generate reactive lipid radicals or catalyze the decomposition of lipid hydroperoxides [3]. This ultimately leads to the formation of off-flavors and aromas. Metal-chelating antioxidants, such as ethylenediaminetetraacetic acid (EDTA), are commonly used in the food industry to prevent metal-catalyzed oxidation, particularly in dressings and mayonnaises. However, due to health concerns and increasing consumer demand for plant-based, natural ingredients and clean-label products, the search for alternative, effective metal chelators has gained significance [4,5]. Proteins and peptides are examples of natural biomolecules gaining traction as potential antioxidants based on their ability to scavenge free radicals, chelate metal ions, and/or inactivate reactive oxygen species [6,7,8,9]. Shorter peptides (2–20 amino acids (AAs)) tend to be more bioactive than longer peptides [7,10].

The world population is projected to reach between 9.4 and 10.1 billion in 2050. This undoubtedly puts pressure on food production to keep up with the increasing demand for nutritious food. Additionally, the impact of food production and agriculture on climate change cannot be neglected. According to the United Nations Environment Program, 17% of global food production is wasted, and between 8 and 10% of global greenhouse gas emissions are associated with food that is not consumed [11]. As reflected in the United Nations Sustainable Development Goal 12, it is essential to rethink production in order to ensure sustainable use of natural resources and reduce food waste. One way to do this is by utilizing side-streams from food production to generate high-value ingredients, such as antioxidants. Potato is the most important non-cereal food crop with a global production of 381 million tons per year [12]. In potato starch production, protein is a valuable by-product due to it being a non-allergenic source of essential AAs [13]. Furthermore, it can be hydrolyzed to produce antioxidant peptides [14].

When screening new antioxidants for their metal-chelating activity, the choice is often to use time-consuming storage experiments in simple food systems where the potential ingredient is added alongside ferrous iron that catalyzes the oxidation [14]. Recently, Surface Plasmon Resonance (SPR) has been used to screen hydrolysates for the presence of metal-chelating peptides [15,16]. SPR is a sensitive, optical technique used to determine the equilibrium dissociation constant (*K*_D_) of a complex formed between an immobilized molecule (e.g., Ni^2+^) and an analyte in solution (e.g., peptide). In this study, Ni^2+^ is used as a proxy for Fe^2+^ due to the similarities between the two. They carry the same charge and have the same coordination number. They have similar electronegativities and are both considered intermediate Lewis acids in Hard and Soft Acid and Base (HSAB) theory [17].

While bulk screening and characterization of hydrolysates provides an excellent tool for evaluating physicochemical properties, it does not provide deeper insight into which peptides within the hydrolysate are in fact responsible for the observed activity. However, the recent development of mass spectrometry (MS)-based peptidomics and subsequent deep learning-based functional predictions has provided tools for obtaining such insight [14,18,19]. Similar approaches have also been demonstrated for other hydrolysate functionalities, such as emulsification, where it has been possible to link quantitative peptide-level composition with bulk functionality [20,21]. While still in its infancy, such a bottom-up approach can therefore provide the missing link between peptide composition of hydrolysates and their functionality, ultimately opening the possibilities of designing targeted hydrolysis strategies.

The aim of this study was to evaluate the potential of SPR as a screening technique for potato protein hydrolysates (PPHs) as metal-chelating antioxidants. More specifically, the goal was to correlate the *K*_D_ of the hydrolysates, determined by SPR, with the development of different oxidation markers obtained during a storage experiment where the PPHs were added as antioxidants to 5% fish oil-in-water emulsions at pH 7. Additionally, the effect of using different combinations of enzymes on antioxidant activity of the hydrolysates was investigated, and peptidomics was applied to characterize the composition of the hydrolysates. With the aim of achieving shorter peptides through a high degree of hydrolysis (DH%), this study utilized sequential hydrolysis, where trypsin (Try) or alcalase (Alc) was paired with Flavourzyme (Fla). Owing to its broad specificity as a heterogeneous combination of various endo- and exopeptidases [22], Flavourzyme is known to facilitate high DH% and the production of short peptides [23]. However, it has been reported that when used solely with a denatured substrate protein of limited solubility, Flavourzyme can yield a significantly lower DH% than anticipated [21]. Consequently, this study was designed to include substrate pre-digestion prior to Flavourzyme addition.

## 2. Materials and Methods

### 2.1. Materials

A high-quality protein concentrate, Protafy 130, extracted from Danish potatoes, was provided by the Danish potato starch and protein manufacturer KMC Amba (Brande, Denmark). This food-grade potato protein concentrate stems from potato fruit juice and is obtained by denaturation of the proteins using heat, precipitation, and subsequently isolation using a decanter centrifuge. The protein is dried in a spin flash dryer and further processed before ending up as Protafy 130 with a declared protein content of a minimum of 80% (KMC provided information).

Alcalase (2.4 L FG-SDS), trypsin (Formea Prime), and Flavourzyme (1000 L) were provided by Novozymes (Bagsværd, Denmark). Cod liver oil was provided by Vesteraalens (Sortland, Norway). The fatty acid (%, *w*/*w*) composition of the fish oil was determined by García-Moreno et al. [24] and was as follows: C14:0 (0.2%), C16:0 (9.4%), C16:1 n-7 (8.6%), C18:0 (2.0%), C18:1 n-9 (16.2%), C18:1 n-7 (4.6%), C18:2 n-6 (1.8%), C18:3 n-3 (0.1%), C20:1 n-9 (12.6%), C20:5 n-3 (9.1%), C22:1 n-11 (5.9%), and C22:6 n-3 (11.1%). The PV of the cod liver oil was measured as 0.28 ± 0.01 meq O_2_/kg oil. All other chemicals and solvents used were of analytical grade.

### 2.2. Production of Hydrolysates

#### 2.2.1. Enzymatic Hydrolysis

Briefly, 3.570 g Protafy 130 was added to a 250 mL blue cap bottle and mixed with distilled water to achieve a protein concentration of 1.9% (*w*/*v*). All samples were placed on a magnetic plate and stirred at 200 rpm for 1 h at ambient temperature. pH was measured using a SI Analytics pH meter (Xylem analytics, Mainz, Germany) and adjusted to pH 8 using 1 M NaOH. Samples were placed in a water bath (Julabo SW22, Julabo USA Inc., Allentown, PA, USA) and heated to 50 °C while shaken at 150 rpm. The enzymatic hydrolysis was initiated by adding the first enzyme (alcalase or trypsin) in a 1% E/S ratio and was run for 1 h with freefalling pH. After completion of the first hydrolysis, all samples were transferred into a separate water bath and heated to 90 °C with a holding time of 15 min to inactivate enzyme. Samples were then cooled to ambient temperature, and pH was adjusted to 7 using 1 M HCl or 1 M NaOH. The enzymatic hydrolysis process was repeated using Flavourzyme (1% E/S ratio) followed by enzyme inactivation. Finally, samples were cooled to ambient temperature, and pH was measured.

Samples were centrifuged in a Sorvall RC6+ (ThermoFisher Scientific, Rockford, IL, USA) for 20 min at 10,000× *g* and 20 °C to collect the supernatant. The weight of the supernatant and pellet was measured. 

#### 2.2.2. Determination of Protein Content Using DUMAS

The protein content of the supernatant was measured in duplicates using the DUMAS method. Briefly, 2.5 g of liquid sample was weighed in a crucible, which was placed in the Rapid MAX N (Elementar Analysensysteme GmbH, Langesebold, Germany). A protein factor of 6.25 was used for converting nitrogen content into protein content. The protein content in the supernatant was then calculated as shown in Equation (1), and the protein yield was calculated as shown in Equation (2).
(1)$${\mathrm{Protein}}_{\mathrm{supernatant}\;}\left(g\right)=\frac{{\mathrm{Protein}}_{\mathrm{dumas}}\;\left(\%\right)}{100\%}\cdot \mathrm{supernatant}\;\left(g\right)$$
(2)$${\mathrm{Yield}}_{\mathrm{protein}}\;\left(\%\right)=\frac{{\mathrm{Protein}}_{\mathrm{supernatant}}\;\left(g\right)}{\mathrm{Protafy}\;\left(g\right)}\cdot 100\%$$

#### 2.2.3. Determination of the Degree of Hydrolysis

The degree of hydrolysis (DH%), defined as the percentage of cleaved peptide bonds in a hydrolysate, was evaluated using the O-phthaldialdehyde (OPA) assay. 

Here, 10 mL of 0.15 M Na_2_CO_3_•10H_2_O was transferred to a 100 mL measuring flask, and 10 mL of 0.6 M NaHCO_3_ and 88 mg of dithiothreitol (DTT) were added. In a 10 mL beaker, 80 mg OPA was weighed out, and 2 mL 96% ethanol was added. The solution was dissolved on a magnetic stirrer in darkness. When dissolved, 10 mL of 1% sodium dodecyl sulfate (SDS) was added. The OPA solution was added to the measuring flask containing DTT, and Milli-Q water was added to the 100 mL mark. The final OPA reagent was kept in the dark until use.

The protein hydrolysate samples were diluted using Milli-Q water to reach a protein concentration between 0.05 and 0.25%. Subsequently, four solutions with dilution factors (DF) of 1, 2, 4, and 8 were made. In duplicate, 20 μL of each dilution was transferred to an Eppendorf MTP 96 microplate, and 200 μL OPA reagent was added to each vial and shaken at 500 rpm in an Eppendorf ThermoMixer C (Thermo Fisher Scientific, Waltham, MA, USA) for 20 s. The absorbance was measured at 340 nm using a microplate reader (Agilent Technologies, Santa Clara, CA, USA).

An 8-point L-serine calibration curve was made using 2-fold dilutions of a 0.5 mg/mL L-serine stock solution. The sample equivalent serine was calculated as shown in Equation (3).
(3)$$“\mathrm{Sample}\;\left(\mathrm{mg}\;\mathrm{Ser}/\mathrm{mL}\right)”=\frac{\left({\mathrm{Abs}}_{\mathrm{sample}}-{\;\mathrm{Abs}}_{\mathrm{blank}}\right)-\;\mathrm{intercept}}{\mathrm{slope}}\cdot DF$$

Abs_sample_ is the absorbance of the sample, Abs_blank_ is the absorbance of the blank, *DF* is the dilution factor, and intercept and slope are obtained from the L-serine calibration curve.

DH% is then calculated as shown in Equation (4).
(4)$$\mathrm{Sample}\;\mathrm{DH}\left(\%\right)=\frac{\mathrm{Sample}\;\left(\mathrm{mg}\;\mathrm{Ser}/\mathrm{mL}\right)}{P\;\cdot 10}\cdot 100\%$$

P is the protein content in percentage. Each dilution was measured in duplicate. 

### 2.3. Fractionation

Four replicates of alcalase and Flavourzyme (AlcFla) and trypsin and Flavourzyme (TryFla) were each merged into one sample. Then, 125 mL of AlcFla and TryFla was transferred to two 250 mL blue cap bottles, serving as unfractionated hydrolysate samples. Samples were kept at −20 °C until further use.

#### 2.3.1. Ultrafiltration

Hydrolysate solutions were fractionated by ultrafiltration using a 300 mL ultrafiltration cell (Millipore, NH, USA) and 5 kDa, 3 kDa, and 1 kDa 76 mm Ultracel^®^ ultrafiltration membranes (Millipore, Jaffrey, NH, USA). Samples were passed through the 5 kDa membrane using a pressure of 5 bars. The permeate was subjected to subsequent fractionations using 3 kDa and 1 kDa membranes. Collecting permeates and retentate resulted in three fractions for each hydrolysate: <1 kDa, 1–3 kDa, and 3–5 kDa. The six fractions, plus the two unfractionated, were lyophilized and stored at −20 °C until further analysis.

#### 2.3.2. SDS-PAGE

SDS-PAGE analysis was done as previously described by Gregersen Echers, Abdul-Khalek, et al. [20]. Briefly, samples were analyzed using pre-casted 4–20% gradient Bis-Tris (Genscript, Piscataway Township, NJ, USA) in a Tris-and 3-(N-morpholino)propanesulfonic acid (MOPS) buffer system according to manufacturer guidelines. PIERCE Unstained Protein MW Marker (ThermoFisher Scientific, Waltham, MA, USA, P/N 26610) was used as marker, and gels were stained using Coomassie Brilliant Blue G250 (Sigma-Aldrich, Taufkirchen, Germany). Visualization was performed by imaging using a ChemDoc MP Imaging System (Bio-Rad, Hercules, CA, USA).

### 2.4. Peptidomics Analysis by nLC-MS/MS

Samples were reduced, alkylated, and desalted as described by Jafarpour et al. [19]. The peptide solutions were loaded equally to 1 μg of protein (based on protein concentration using Nanodrop A280 (Thermo Scientific, Karlsruhe, Germany)). Peptides were separated over a 120 min gradient from 95% solvent A (0.1% aq. Formic acid (VWR, Søborg, Denmark)) to 100% solvent B (0.1% formic acid in 80% aq. Acetonitrile (FischerScientific, Karlsruhe, Germany)) using an EASYnLC 1200 system (Thermo Scientific) equipped with a reverse phase Acclaim Pepmap RSLC analytical column (C18, 100 Å, 74 μm × 50 cm (Thermo Scientific)). Eluted peptides were analyzed on a Q Exactive HF mass spectrometer (Thermo Scientific) using a top 20 data-dependent method. Survey scans were performed from 200 *m*/*z* to 3000 *m*/*z*, and maximum ion injection time was 50 ms and 100 ms for MS and MS/MS scans, respectively. All other settings were identical to those in a previous report [21].

#### 2.4.1. LC-MS/MS Data Analysis

Raw data from LC-MS/MS were initially analyzed using MaxQuant v.2.2.0.0 as previously described [21], with slight modifications. Briefly, data were searched against a curated version of the full *Solanum tuberosum* (tax:4113) protein database from UniProt, as previously described. Analysis was performed using unspecific in silico digestion to identify peptides in the range of 3 to 65 AAs. Standard settings were applied, including common contaminants and reverse sequences for false discovery rate (FDR) control, using a 10% FDR on both the peptide and protein levels. The mass spectrometry proteomics data have been deposited to the ProteomeXchange Consortium PRIDE partner repository with the dataset identifier PXD050312 and 10.6019/PXD050312.

#### 2.4.2. Processing of MaxQuant Data and Prediction of Antioxidant Properties

Unambiguous contaminants and reverse hits were initially removed, whereafter the intensity-weighted relative peptide abundance estimation methodology [25] was employed to estimate sample-level peptide length distribution, charge distribution, and molar AA abundance. The antioxidant potential of identified peptides was predicted in silico using AnOxPrePred [18]. The algorithm was operated in “Peptide Mode” and allowed peptides of 3–30 AAs in length for prediction of free radical-scavenging (FRS) and metal-chelating (CHE) potential. For quantitative comparison, intensity-weighted peptide abundance was summed for peptides with scores above the algorithm threshold (FRS ≥ 0.43 and CHE ≥ 0.30) [19]. Score distributions were analyzed by pairwise comparison using one-way ANOVA and Tukey and subsequently visualized using GraphPad Prism (v.9.3.1). Overlap of peptide identifications between fractions was investigated and visualized using UpSet plots through SRPLOT (http://www.bioinformatics.com.cn/srplot, accessed on 4 July 2023) as well as Venn diagrams using jvenn.

### 2.5. Surface Plasmon Resonance

#### 2.5.1. Glycine Equivalence as Determined by OPA

Peptide concentrations were quantified using OPA assay and expressed in mM glycine equivalents. The method was adapted from Canabady-Rochelle et al. [15]. The OPA solution was prepared in the following way: 40 mg OPA reagent was dissolved in 1 mL methanol, and 100 mg N,N-dimethylmercaptoethylammonium was dissolved in 5 mL Borax buffer (100 mM sodium tetraborate, 1 *w*/*v* sodium dodecyl sulfate, pH 9.3). The two solutions were mixed in a volumetric flask, and Borax buffer was added up to 50 mL. For the glycine calibration curve, a stock solution of 5 mM glycine in water was made. The stock solution was diluted to 4, 3, 2, 1.5, 1, 0.75, 0.5, and 0.25 mM. PPHs were dissolved in water at 1 mg/mL. Then, 20 µL of the hydrolysate solution was mixed with 200 µL of OPA solution in the wells of a 96-well plate in triplicate. The plate was shaken for 20 s at 500 rpm in an Eppendorf ThermoMixer C (Thermo Fisher Scientific, Waltham, MA, USA) followed by incubation for 3 min at room temperature in the dark. The absorbance was read at 340 nm using a BioTek EON Microplate Spectrophotometer (Agilent Technologies, Santa Clara, CA, USA). A blank with 20 µL water was included, and the absorbance was subtracted from the sample absorbance. The glycine equivalent of each hydrolysate solution was calculated using the glycine calibration curve and expressed as mM Eq. glycine ± SD.

#### 2.5.2. SPR

The affinity between the peptides and immobilized Ni^2+^ was analyzed by surface plasmon resonance (SPR). The analysis was carried out on a Biacore X100 instrument (Cytiva, Uppsala, Sweden) equipped with a nitriloacetic acid (NTA) sensor chip at 25 °C according to Canabady-Rochelle et al. [15] and adapted from Knecht et al. [26]. Binding experiments were performed at a flow rate of 20 µL min^−1^. Ni^2+^ was loaded onto the NTA chip using a 0.5 mM NiCl_2_ solution for 1 min followed by a 1 min stabilization period. A NTA flow cell without Ni^2+^ was used as the reference. The running buffer was PBS1X at pH 7.4 (6.7 mM Na_2_HPO_4_·2H_2_O, 0.125 mM KH_2_PO_4_, 15 mM NaCl, 0.005% Tween 20). PPHs were dissolved in freshly prepared PBS1X at seven different concentrations (0.25 to 10 mM Eq. Gly) before each experiment. The goal was to obtain a hyperbolic profile and reach saturation of R_max_. Each concentration of hydrolysate was injected into both flow paths for 270 s followed by 270 s of undisturbed dissociation time. Between each investigated concentration, the chip surface was regenerated with a 500 mM imidazole solution followed by a regeneration solution (3 mM EDTA in PBS1X). Regeneration of the chip was carried out by 1 min injection of imidazole solution followed by a washing step with 350 mM EDTA solution. The surface was then washed with a SDS solution (0.5% *v*/*v*) for 1 min at 40 µL min^−1^ followed by an extra wash with running buffer. Each regeneration and washing step were repeated twice. One buffer blank before and after each sample series was used for double referencing during data processing. Duplicates were made of the 1st, 4th, and 7th concentrations. 

The obtained sensorgrams were processed with BIAevaluate software (version 2.0.2 Plus Package) from Cytiva (Uppsala, Sweden) to get the sorption isotherms (resonance unit (RU), corrected by the offset value, plotted as a function of the concentration of hydrolysate). For each experiment, the response from the blank run was subtracted, and the equilibrium dissociation constant (*K*_D_, M) was determined by fitting the experimental data with the 1:1 binding model [15].

### 2.6. Emulsion Production and Storage Experiment

Briefly, 220 g of 5% cod oil-in-water emulsions stabilized with 1 wt% TW20 were produced as described by Yesiltas et al. [14] using 0.05 wt% hydrolysates as the antioxidant instead of the synthetic peptides used in that study. EDTA was included as a positive control at 0.0075 wt%, which is the maximum allowed concentration used in mayonnaise in the EU. A negative control without any antioxidants was included as well. Samples were collected for physical characterization and oxidative stability analyses during 9 days of storage at room temperature in darkness. 

### 2.7. Physical Stability of Emulsions

#### 2.7.1. Droplet Size Distribution

Droplet size was measured on day 1 and day 9 with laser diffraction using Mastersizer 2000 (Malvern Instruments, Ltd., Worcestershire, UK). Emulsions were diluted in recirculating water at 3000 rpm until reaching an obscuration between 12 and 15%. The refractive indices of sunflower oil (1.469) and water (1.330) were used for particle and dispersant, respectively. Measurements were done in duplicate and given as the mean diameters of the volume weighted (D[3, 2]) and the surface weighted (D[4, 3]).

#### 2.7.2. Zeta Potential

Zeta potential was measured on day 1 using Zetasizer Nano ZS (Malvern Instruments Ltd., Malvern, UK). Samples were prepared by diluting 80 μL of emulsion in 40 mL 10 mM sodium acetate—10 mM imidazole buffer. Measurements were done in duplicate using a DTS-1070 disposable folded capillary cell (Malvern Instruments, Ltd., Malvern, UK) in the zeta potential range of (−) 100 to (+) 50 mV.

### 2.8. Oxidative Stability

Oxidative stability analyses were done on the samples collected on days 0, 3, 6, and 9 during the storage experiment. Samples were purged with nitrogen and stored at −40 °C until analysis.

#### 2.8.1. Oil Extraction and Peroxide Value

Extraction of oil from emulsions was done based on the method described by Bligh & Dyer [27] using a reduced amount of chloroform/methanol (1:1, *w*/*w*). Two extractions were made from each emulsion. Peroxide value (PV) was determined for the oil extracts using the spectrophotometric ferric-thiocyanate method at 500 nm according to Shantha & Decker [28] with a Shimadzu UV-1280 spectrophotometer (Holm&Halby, Brøndby, Denmark). Measurements were made in duplicate.

#### 2.8.2. Determination of Tocopherol Content

Tocopherol content was determined for the Bligh and Dyer extracts using HPLC as described by Yesiltas et al. [14].

#### 2.8.3. Determination of Secondary, Volatile Oxidation Products Using Dynamic Headspace GC-MS

Determination of secondary volatile oxidation products was done using dynamic headspace GC-MS as described by Yesiltas et al. [14]. Briefly, 30 mg internal standard (4-methyl-1-pentanol) and 5 mL water were added to 4 g of emulsion in a purge bottle. The purge bottle was heated in a water bath at 45 °C for 30 min under purging with nitrogen (150 mL/min), and the volatile compounds were trapped in Tenax GR tubes. The trapped volatiles were desorbed using an automatic thermal desorber (TurboMatrix 650 ATD, Perkin Elmer, Norwalk, CN, USA) connected to a gas chromatograph (Agilent 6890 N, Palo Alto, CA, USA; Column: DB-1701, 30 m × 0.25 mm × 1.0 μm) using helium gas flow (1.3 mL/min). The temperature program was as follows: 45 °C for 5 min, 1.5 °C/min from 35 to 55 °C, 2.5 °C/min from 55 to 90 °C, 12 °C/min from 90 to 220 °C, and held for 4 min at 220 °C. The volatile compounds were analyzed by electron ionization mass spectrometry (EI-MS) (Agilent 5973, Agilent Technologies, USA) at 70 eV. The EI-MS was operated in scan mode (from 30 *m*/*z* to 250 *m*/*z*) and identified by searching fragmentation spectra against EI-MS spectral libraries (Wiley 138 K, John Wiley and Sons, Hewlett-Packard). Measurements were done in triplicate. The selected standards were 3-methyl-butanal, 2-ethyl furan, pentanal, 1-penten-3-ol, hexanal, heptanal, (E)-2-heptenal, 1-octen-3-ol, benzaldehyde, octanal, (E,E)-2,4-decadienal, and (E,E)-2,4-heptedienal. A stock solution of the standards was made in ethanol and diluted into 7 concentrations (0.5–100 µg/mL) for the calibration curve. Calibration samples were then prepared in triplicate as described for the emulsion samples with 30 mg volatile standard solution replacing the 30 mg internal standard.

### 2.9. Statistical Analysis

Statistical analysis was done using Statgraphics 18 (Statistical Graphics Corp., Rockville, MD, USA). A multiple sample comparison was performed to identify the significant differences between samples at certain sampling days and between sampling days for each sample during storage using Tukey as a post-hoc test at a *p* < 0.05 significance level. Data are expressed as mean ± standard deviation.

## 3. Results and Discussion

### 3.1. Production of Hydrolysates

#### 3.1.1. Enzymatic Hydrolysis

The protein content of Protafy 130 was initially determined to be 84% (*w*/*w*). Using this as a reference, the protein yield after hydrolysis ranged from 27.2 ± 0.1% to 40.2 ± 4.2%, and the DH% ranged from 13.1% to 38.8% (Table 1).

A significant difference was observed in DH% between Alc and Try. The lower DH% of Try may be due to the high specificity of trypsin; oppositely, the higher DH% of Alc may be due to the broad selectivity of alcalase [29]. The DH% seems to be correlated with both protein yield and protein content, which indicates that hydrolysis plays an important role in solubilizing a denatured potato protein concentrate such as Protafy 130. The same tendency has been observed in other studies hydrolyzing potato protein [21,30]. Significant increases in DH% and protein yield were observed when using a sequential combination of two enzymes compared to using an endoprotease alone. Compared to Alc and Try, DH% increased significantly for AlcFla and TryFla from 20.7 ± 0.7% to 38.8 ± 3.6% and 13.1 ± 1% to 38.1 ± 3%, respectively. This demonstrates that the endopeptidases, which work within the proteins, facilitate the work for the exopeptidases by increasing cleavage site accessibility. Due to different ways of calculating DH%, differences in protein source, and different hydrolysis conditions, it is difficult to compare with other studies. However, an increase in DH% using Flavourzyme as a second enzyme has also been reported for lentil, soy, sunflower seed, and rapeseed protein [31,32]. Moreover, as increased DH% could be correlated to increased antioxidant activity of peptides [33], the AlcFla and TryFla hydrolysates were chosen for further evaluation.

#### 3.1.2. Size Fractionation

AlcFla and TryFla were subjected to ultrafiltration to obtain three different size fractions (<1 kDa, 1–3 kDa, 3–5 kDa). An unfractionated sample (AFPPH and TFPPH) was included for both hydrolysates as reference. The eight fractions (AF1, AF13, AF35, AFPPH, TF1, TF13, TF35, and TFPPH) were freeze-dried. Only 40 mg of TF1 was obtained, and the sample was very sticky, which may be due to insufficient freeze drying. This has also previously been reported for dried peptides and hydrolysates [34], indicating that the phenomenon may also be related to residual carbohydrates or the physicochemical properties, particularly water-holding capacity, and the hygroscopic nature of this particular fraction [35]. Based on this (particularly the low yield), it was decided not to continue with TF1. The protein content differed significantly between the fractions of PPHs. AF35 had the highest protein content of 75.3 ± 1.24%, whereas AF1 had the lowest protein content of 62.9 ± 0.2% (Appendix A).

SDS-PAGE analysis (Appendix A) confirmed that the PPHs and the fractions all contained peptides of lower molecular weight, as indicated by the smears at the bottom of the lanes. For AFPPH and TFPPH, visible bands can be observed at 25 kDa, corresponding to the mass of the enzymes, which were efficiently removed by ultrafiltration. The results of AFPPH and TFPPH looked similar, corresponding with the similar DH% (Table 1). As the intended mass range of fractions decreased, a downshift in the molecular weight (MW) of the smears was observed, indicating that stepwise ultrafiltration was capable of fractionating the hydrolysates. For AF1 and TF13, no clear smears are visible, indicating little or no presence of higher MW peptides. In this respect, it should be kept in mind that the band response is mass dependent [36]. This means that even though only faint staining is visible, a high quantity of low MW peptides can still be present, as these may run off the gel without giving rise to a response and, more importantly, are less confined within the gel matrix during de-staining without explicit fixation. While loaded in the same amount on the gel, a smear in the same mass range is more visible for TF35 compared to AF35, indicating that TF35 could have an increased content of higher MW peptides.

### 3.2. Peptidomics

Peptidomics was used to identify the peptides present in hydrolysate fractions in a semi-quantitative manner. In addition to peptide sequences and their relative abundance, information was obtained regarding the AA composition as well as the size and charge distribution of the peptides, allowing for determination of summary statistics as descriptors of the average state of individual fractions. Across the seven fractions, 12,047 peptides were identified (Appendix A) following filtering of false positives and unambiguous contaminant peptides (836 IDs). All fractions had a high number of peptide IDs ranging from ~4000 (AF1) to ~6300 (TFPPH), where unfractionated hydrolysates had the highest number of identifications and higher MW fractions had more identifications than lower MW fractions. While previous studies on potato protein hydrolysates using single-enzyme digestion have identified a substantially higher number of peptides, the sequential approach used here results in much higher DH%. This, in turn, decreases average peptide length, thereby decreasing the combinatorial space, and furthermore increases the relative content of dipeptides and single AAs, which are not compatible with the applied workflow. The weighted average peptide length shows that all fractions generally had a low average peptide length and slightly negative net charge at pH 7 (Table 2). Generally, the weighted mean charge of the fractions increased slightly towards zero when decreasing the intended mass range. The low average length is further substantiated by the high abundance of peptides identified in the 3–20 AA range (Appendix A) and in agreement with the extensive hydrolysis and high DH% obtained by the sequential approach. The average length generally decreased when decreasing the intended mass range of fractions, but not to the same extent as would be expected, since both the arithmetic and weighted means of the larger MW fraction (3–5 kDa) were below 10 kDa, indicating that fractionation was not fully successful. Furthermore, a substantial number of shorter peptides were still present in the higher MW fractions. This observation relates to the endogenous length-bias obtained when using an experimental setup typically employed for bottom-up proteomics and is also in line with previous studies using the same approach. Nevertheless, the differences in average length did indicate the desired effect from ultrafiltration and corresponded with differences observed in SDS-PAGE (Appendix A).

AF1 had the shortest average peptide length, and the weighted distribution was dominated by peptides with 3–5 AAs (Appendix A). Alcalase generally resulted in shorter peptides compared to trypsin, likely due to the broader specificity and thus greater extent of hydrolysis for alcalase compared to trypsin, which only cleaves at the C-terminal of Arg and Lys. This was unexpected since the DH% was not significantly different between the two hydrolysates (Table 1). However, as trypsin cleaves C-terminally of Arg/Lys, this may reflect a larger relative proportion of alkaline residues, which may result in bias for amine-specific assays, such as OPA. Moreover, inclusion of alkaline residues, particularly Lys, has been shown to increase the peptide intensity in MS analysis [24,37], which may ultimately affect the determined mean using an intensity-weighted approach. Indeed, a higher relative proportion of Lys was observed in trypsin-treated samples (Table 2). From the analysis of AA composition, there are also other notable differences between PPHs and their fractions. When compared to trypsin-treated samples, alcalase-treated samples, particularly the low MW fractions AF1 and to some extent AF13, appear to be somewhat depleted of peptides containing AAs such as Arg, Glu, Gly, Ile, Lys, Ser, and Thr. In contrast, these fractions were found to be rich in peptides comprising aromatic AAs (Phe, Trp, Tyr) and Leu. Enrichment and depletion of these AAs may be favorable, as this correlate well with previous reports on AA composition in antioxidant peptides [18]. The specific AA composition may therefore result in different bioactive properties despite comparable DH% as well as length and charge distributions.

#### Prediction of Peptide-Level Antioxidant Properties

Identified peptides were analyzed using AnOxPePred [18], which is a deep learning-based bioinformatics tool used to predict the probability of peptides being free radical scavengers (FRS) or metal chelators (CHE) based on their AA sequence. All hydrolysates were predicted to contain a considerably higher number of peptides with FRS scores over the threshold value (Figure 1A) compared to peptides with CHE scores above the threshold (Figure 1B). Considering the score distribution, only subtle differences were observed. The only noticeable difference is a slight upshift in the arithmetic mean and upper quartile for AF1 in both FRS and CHE scores, while a slight downshift was observed for TF35 and TFPPH in relation to CHE scores. While these differences are indeed small, AF1 had a significantly higher mean score (adjusted *p* value < 0.0001) than all other fractions for both FRS and CHE (Appendix A). These differences become much more apparent when considering the quantitative aspects of the peptidomics analysis, where the proportion of peptides with a FRS score above the threshold increased substantially for AF1 (47%) compared to the remaining fractions (26–33%). For CHE scores, the differentiation is also more evident as AF1 comprises peptides scoring over the threshold score accounting for 15% of the total peptide-level intensity, while TF35 and TFPPH contained substantially less (3.6% and 3.9%, respectively) and also less than remaining fractions (5.0–6.8%). Overall, the content of peptides with scores above threshold values appeared to increase when decreasing the intended mass range of the fractions.

While the identified peptides differed substantially across all fractions (Figure 1C and Appendix A), a substantial number (1196) was conserved, corresponding to a relative share of all peptide IDs from 19% (TFPPH) to 30% (AF1). The number of conserved peptides increased substantially when considering the PPHs and theirs fractions separately. Specifically, 1971 peptides (33–50%) were conserved between AF fractions, while 3467 peptides (55–66%) were conserved between TF fractions (Appendix A). The increased conservation in TF fractions also reflects the higher specificity of trypsin compared to alcalase. The largest number of peptides found uniquely in one fraction (Figure 1C) was identified for AF1 (633) corresponding to 16% of all peptide IDs in the fraction, followed by AFPPH (410, 6.7%) and TFPPH (402, 6.4%). This could indicate that the large number of unique peptides found in AF1 may explain why this fraction generates higher scores than the other fractions. This subset represents 12.1% of the total peptide intensity for AF1 and furthermore has a mean CHE score of 0.26 (data not shown), which is higher than the arithmetic means generally observed (~0.23) for the fractions (Figure 1B). Overall, peptidomics and bioinformatic analyses suggest that AF1 should have the largest potential for not only acting as a metal chelator but also as a radical scavenger and thus have higher potential for improving oxidative stability in e.g., emulsions, while TF35 and TFPPH may display lower activity.

### 3.3. Surface Plasmon Resonance (SPR)

The binding between peptides in solution and Ni^2+^, which is partially complexed by immobilized nitrilotriacetic acid (NTA) at pH 7.4, was investigated by SPR. Ni^2+^ was chosen due to its similarities with Fe^2+^, as the iron ion is the most problematic in terms of catalyzing lipid oxidation.

In SPR, the equilibrium dissociation constant (KD) is determined by mathematically fitting the equilibrium response at different concentrations according to a 1:1 binding model. When using NTA as the capture molecule, there are two coordination sites remaining on the nickel ion for the peptide to bind to. This must be considered when comparing the SPR data with metals in solution. The sorption isotherms obtained in this study are shown in Appendix A. For all samples, the response (RU) was concentration dependent, indicating the presence of peptides with affinity for Ni^2+^ and thus most likely also with affinity for Fe^2+^. The sorption isotherms had a hyperbolic profile with a saturation plateau or a tendency for saturation.

The obtained dissociation constants are shown in Table 3. Interestingly, the KD values were all of the same order of magnitude regardless of enzyme treatment or size fractionation. For AFPPH, TF35, and TFPPH, the obtained KDs were greater than half of the maximum concentration used. This indicates that higher concentrations are required to accurately determine the true KD, suggesting that the current values are likely underestimations. Given these limitations and the high standard errors, it is more reliable to consider the range of peptide affinities rather than specific values. Consequently, a quantitative comparison of the different hydrolysates based on their KD values is not feasible.

In their study, El Hajj et al. [16] also observed similar patterns in the sorption isotherms of their samples, with results indicating either the achievement of saturation or a trend towards it, mirroring the tendencies seen in our findings. They used the same experimental setup and studied the effect of different enzymes on <1 kDa size fractionated soy and pea protein hydrolysates. They found no correlation between DH% and the affinity of the hydrolysates. Additionally, their results indicated that alcalase typically yielded lower KD values compared to Protamex.

Nevertheless, it is noteworthy that in the present study AF1 obtained the lowest KD, thereby following the predicted activity based on CHE scores. Overall, the obtained KD seemed to follow the same trends as predicted CHE scores, where lower MW fractions displayed higher affinity than larger MW fractions and AF fractions had higher affinity than the TF fractions in the same MW range.

When analyzing SPR data, it is important to recognize the inherent challenges due to the dynamic and flow-based nature of this assay. Specifically, when working with complex mixtures like hydrolysates, SPR essentially becomes a pseudo-competition assay. In such a setup, peptides with higher binding affinities may displace those with lower affinities, a phenomenon less evident in simpler assay conditions. This dynamic can complicate the achievement of a definitive saturation level or plateau in SPR, making it challenging to compare with single-compound systems.

### 3.4. Storage Experiment with Emulsions

In the storage experiment, 110 mg of PPHF was added to emulsions. The results of the measured protein content showed that even though the same amount of PPHF was added to each emulsion, the emulsions did not contain the same quantity of peptides. The calculated actual amount showed that the least amount of protein added was 69.1 mg (AF1) and the largest amount was 82.9 mg (AF35). The different protein contents must be considered when comparing the oxidative and physical stability of the emulsions.

#### 3.4.1. Physical Stability of the Emulsions

The physical stability of the emulsions during the 9 days of storage was evaluated to assess if the fractionated and unfractionated potato protein hydrolysate had a negative impact on the emulsion stability. No creaming was observed visually throughout the 9 days of storage, indicating stable emulsions.

##### Droplet Size Distribution

Emulsions stabilized with 1% wt Tween20 had similar D[3, 2] and D[4, 3] values on day 1 ranging from 0.118 to 0.121 and 0.180 to 0.191 µm, respectively (Appendix A). The emulsions were stable during the storage experiment, and no considerable change in mean droplet size was observed after 9 days with values ranging from 0.118 to 0.123 and 0.181 to 194 µm for D[3, 2] and D[4, 3], respectively. Although mean droplet sizes were similar, significant differences were observed between emulsions, which is mainly due to the low standard deviations. Yesiltas et al. [14] reported similar mean droplet size values for emulsions produced in the same way and stabilized with Tween20 (0.121–0.132 and 0.188–0.229 µm for D[3, 2] and D[4, 3], respectively). The results show that the peptides in the hydrolysates and fractions hereof did not affect the droplet size in the emulsions during storage.

##### Zeta Potential

All emulsions exhibited absolute zeta potential values below 20 mV, ranging from −18.43 ± 3.78 to −12.38 ± 1.02 mV (Appendix A). These values suggest that electrostatic interactions in the emulsions were not entirely effective in providing the necessary electrostatic repulsion to prevent droplet flocculation [38]. However, previous studies have proposed that the low absolute zeta potential can be attributed to the non-ionic nature of Tween20, resulting in limited electrostatic repulsion between oil droplets and subsequently low zeta potentials in otherwise stable emulsions [14,39]. The control and EDTA samples displayed higher absolute zeta potentials compared to the emulsions containing PPHFs, although the differences were not statistically significant. This suggests that the presence of peptides in the PPHFs had minimal or negligible influence on the electrostatic interactions in the emulsions.

#### 3.4.2. Oxidative Stability of Emulsions

##### Peroxide Value

A similar trend in peroxide development was observed across all emulsions, with the PV increasing significantly from day 0 to day 6 with no observable lag phase as shown in Figure 2. It is noteworthy that all emulsions had high PV on the day of production, ranging from 4.5 ± 0.1 to 8.4 ± 0.3 meq O_2_/kg oil, when compared to the PV of the fresh fish oil (0.28 ± 0.01 meq O_2_/kg oil). This finding is consistent with a study by García-Moreno et al. [24], who also observed a high PV on the production day of 5% oil-in-water emulsions at pH 7 with synthetic peptides as emulsifiers. A significantly lower PV on day 0 was observed for the EDTA emulsion compared to all other emulsions in the present study. This could be attributed to the strong metal-chelating activity of EDTA, which can (i) prevent the metal-catalyzed initiation of autoxidation or (ii) prevent the metal-catalyzed breakdown of endogenous hydroperoxides into peroxyl radicals, which accelerates the propagation step of lipid oxidation.

The almost linear increase of PV from day 0 to day 6 can be attributed to the prooxidative activity of the added iron and a lack of sufficient metal-chelating or free radical-scavenging activity at the early stages of lipid oxidation across all samples. The same tendency was observed by Yesiltas et al. [14] when using synthetic peptides as antioxidants in 5% fish oil-in-water emulsions stabilized with Tween20 during 8 days of storage.

On day 6 (before the decomposition of hydroperoxides), the PV of the negative control (without antioxidants) was significantly higher than all seven hydrolysates, confirming the presence of antioxidant peptides in the hydrolysates. While EDTA only works as a metal chelator, the peptides present in protein hydrolysates could potentially exhibit both chelating and/or FRS activity as shown in previous studies [37,38]. A decrease in PV was observed from days 6 to 9 for all emulsions, except for TF13 and TF35, which can be attributed to the decomposition of hydroperoxides into secondary volatile oxidation products [3]. The negative control showed a more pronounced decrease in PV compared to both hydrolysates and EDTA, which could be due to the lack of metal-chelating agents.

##### Tocopherols

Tocopherols are natural antioxidants present in fish oil. A decrease in tocopherol concentration during storage indicates consumption of tocopherols to counteract lipid oxidation. Due to the order of consumption of the different tocopherol homologues, α-tocopherol was highlighted (Figure 2). The observed trends echo those of the PV results. For the negative control, α-tocopherol was almost depleted after 3 days of storage, while EDTA reduced consumption of α-tocopherol to 21.7% after 9 days (Appendix A). All seven hydrolysates performed worse than EDTA, but significantly better than the negative control with α-tocopherol consumption ranging from 54.3 to 81.5% after 9 days for AF1 and TFPPH, respectively. This indicates antioxidant activity of all PPHFs. The same tendencies were observed by Yesiltas et al. [14] where α-tocopherols were consumed more rapidly in the control emulsion compared to emulsions containing peptides. The best performing peptides exhibited α-tocopherol consumption of roughly 60% after 9 days of storage. Among the emulsions tested in our study, the smaller size fractions, such as AF1, AF13, and TF13, performed significantly better than the larger size fractions and the unfractionated samples after 9 days (54.3–63.7% consumption compared to 71–81.5% consumption of α-tocopherol). This aligns with previous studies [7,10] showing that shorter peptides have higher antioxidant activity.

##### Development of Secondary Volatile Oxidation Products

Ferrous iron facilitates the decomposition of lipid hydroperoxides into secondary volatile oxidation products (SVOP), leading to off-flavors. The formation, or lack of formation, of SVOPs can therefore be used to indicate the presence of metal-chelating peptides in the hydrolysates. In this study, the formation of 12 SVOPs were evaluated during the 9-day storage experiment. Three compounds were selected as representatives based on trend of formation, abundance, or nature of origin. 1-Penten-3-ol represents SVOPs from the decomposition of ω-3 PUFAs [39], hexanal represents decomposition of ω-6 PUFAs, and 3-methyl-butanal represents the two Strecker aldehydes found in the emulsions. The results of the three volatiles are shown in Figure 3. The remaining volatiles are shown in Appendix A. Statistical differences between emulsions and between days are shown in Appendix A.

Overall, the formation of SVOPs aligned with the trends observed for lipid hydroperoxide formation and tocopherol consumption. Except for the two Strecker aldehydes, the control had significantly higher SVOP concentrations throughout the storage period compared to the hydrolysates. This shows that all seven hydrolysates were able to act as antioxidants in simple emulsions, which could, in part, be due to metal-chelating activity of the peptides. However, the hydrolysates generally performed worse than EDTA in suppressing volatile formation. Comparing the formation of 1-penten-3-ol and hexanal, it is clear that the hydrolysates suppressed the formation of hexanal to a greater extent than 1-penten-3-ol and thus behaved more similarly to EDTA for hexanal suppression.

When evaluating the effect of the individual hydrolysates, some trends were observed for 1-penten-3-ol and hexanal: smaller size fractions (<1 and 1–3 kDa) generally performed better than larger ones irrespective of enzyme treatment, although differences were not significant. Interestingly, AF13 was not significantly different from EDTA with respect to hexanal formation at day 9 (*p* < 0.05), hinting at the potential of this particular enzyme/fraction combination. Considering the variation in peptide concentration added to the emulsions, it is noteworthy that a lower amount of pure protein (0.03%) was incorporated into AF1 emulsions compared to the AF13, AF35, and AFPPH emulsions (0.04%). These findings further suggest that the size of the peptides significantly affect antioxidant activity, with shorter peptides appearing more potent. This aligns with previous studies on peptides as antioxidants [7,14,39]. While size is important, other factors such as AA composition and sequence, molecular conformation, overall charge, and hydrophilicity/hydrophobicity all affect the antioxidant activity of peptides [6,39].

3-Methyl-butanal is formed by Strecker degradation of leucine [40] and is often associated with beer due to its malt aroma. While not an unpleasant flavor, the presence of 3-methyl-butanal could contribute to an unwanted aroma profile. It could therefore pose a challenge when utilizing PPHFs as antioxidants in food products. In our study, 3-methyl-butanal was not detected in the control or EDTA emulsions, as no protein was introduced (Figure 3). The presence of 3-methyl-butanal can be explained by the use of Flavourzyme, which contains exopeptidases. Flavourzyme can result in a high degree of hydrolysis, which can potentially explain why leucine may have been present as a free AA.

### 3.5. Can SPR, Peptidomics, and Bioinformatics Be Used as Alternative Screening Methods?

The obtained results from SPR and the storage experiment indicated that there is a relationship between NTA-Ni^2+^ SPR at pH 7.4 and Fe^2+^ chelation in emulsions at pH 7. All seven hydrolysates were found to complex Ni^2+^ using SPR, while also suppressing oxidation in emulsions with added Fe^2+^. Large standard errors in SPR data made direct comparison challenging. However, generally, smaller size fractions demonstrated lower dissociation constants and performed better in the storage experiment. It is interesting to note that the Ni^2+^/EDTA complex in aqueous solution has a *K*_D_ of 4 × 10^−13^ µM [26]. The *K*_D_ values obtained in this study for the hydrolysates were all many orders of magnitude higher than that of EDTA, which means that the hydrolysates had substantially lower affinity for Ni^+^ than EDTA. This comparison must be done with caution as the Ni^2+^/peptide complexes are not free in solution but are facilitated by the immobilized NTA molecule. This reduces the number of coordination sites on the nickel ion from six to two. EDTA is used to regenerate the SPR chip, and it is therefore not possible to analyze EDTA in the same way. Given the large differences in *K*_D_ between the hydrolysates and EDTA, it is interesting that the difference in performance as metal chelators during the storage experiment was not as pronounced. This was especially evident from the formation of hexanal (Figure 3B) where AF13 was not significantly different from EDTA after 9 days of storage. This suggests that the mode of action of the hydrolysates as antioxidants is more complex than just metal chelation. The observed inhibition of oxidation may be attributed to other processes/functions, such as radical scavenging, that a complex hydrolysate is able to exhibit. This was also suggested by the peptidomics results, where all fractions are predicted to contain peptides with chelating and/or scavenging properties. Comparing the PV and SVOP results, the difference between control and the remaining samples are clearer for SVOP than for PV. This further indicates the presence of metal-chelating peptides in the hydrolysates as they can prevent the decomposition of the peroxides to SVOPs.

Across all storage experiments, the lowest MW fraction from alcalase- and Flavourzyme-treated potato protein (AF1) performed better. This correlates with this fraction not only obtaining the lowest *K*_D_, but also that the peptide composition herein was both qualitative and quantitatively predicted to display both higher chelating and radical-scavenging properties. The identified peptides in this fraction were determined to have a substantial enrichment of certain AAs (Leu, Phe, Trp, and Tyr) and depletion of other AAs (Arg, Glu, Gly, Ile, Lys, Ser, and Thr). Overall, these results suggest that not only do shorter peptides tend to display improved antioxidant properties, but the specific AA composition plays a key role. Whether these findings relate to the composition on the individual peptide level or can be generalized to the bulk AA composition is not possible to say at this stage.

## 4. Conclusions

In this study, we investigated the potential of surface plasmon resonance (SPR) as a screening tool for metal-chelating activity in protein hydrolysates. Sequential hydrolysis with alcalase or trypsin in combination with Flavourzyme resulted in high degree of hydrolysis and short peptides with antioxidative properties. The two hydrolysates were fractionated using ultrafiltration, and a combination of peptidomics and bioinformatics allowed for a qualitative and quantitative estimation of the antioxidant potential of the different fractions obtained. All fractions were predicted to contain metal-chelating and radical-scavenging peptides, although one particular low MW fraction (AF1) stood out as having the highest potential as both a metal chelator (15% chelating peptides) and free radical scavenger (47% FRS peptides).

Although SPR showed promise, differentiating fractions based on dissociation constants determined using SPR was challenging due to large standard errors. While SPR may not be suited for ranking hydrolysates by activity, it may be used to determine whether a hydrolysate could contain metal-chelating peptides. This presents a potential alternative to time-consuming storage experiments for initial screening experiments.

In terms of the storage experiments, the hydrolysate fractions suppressed the formation of secondary volatile oxidation products (SVOP) compared to the control without added antioxidants, indicating the presence of peptides with metal-chelating and possibly also radical-scavenging activity. However, they did not suppress formation of SVOPs to the same degree as EDTA. Our findings are consistent with the general understanding that peptide length is a significant factor for antioxidant activity. However, our findings also highlight that length cannot be regarded as a single determinant, as the data emphasize the contribution of factors such as AA composition and sequence order.

In conclusion, our results emphasize the potential of protein hydrolysates as food antioxidants and indicate that both SPR and combined peptidomics/bioinformatics may be viable screening tools to assess the antioxidant potential of hydrolysates. The insights from our SPR analysis underscore the need for advanced data interpretation models capable of accommodating the competitive dynamics observed in complex mixtures like hydrolysates. Future research should focus on addressing the current limitations, e.g., improving the precision of dissociation constants determined by SPR, and conducting experiments with gradually increasing complexity, including displacement studies, to provide a more comprehensive understanding of hydrolysate behavior in SPR assays. Finally, the potential of hydrolysates as antioxidants in other food matrices should be explored.

## Figures and Tables

**Figure 1 antioxidants-13-00346-f001:**
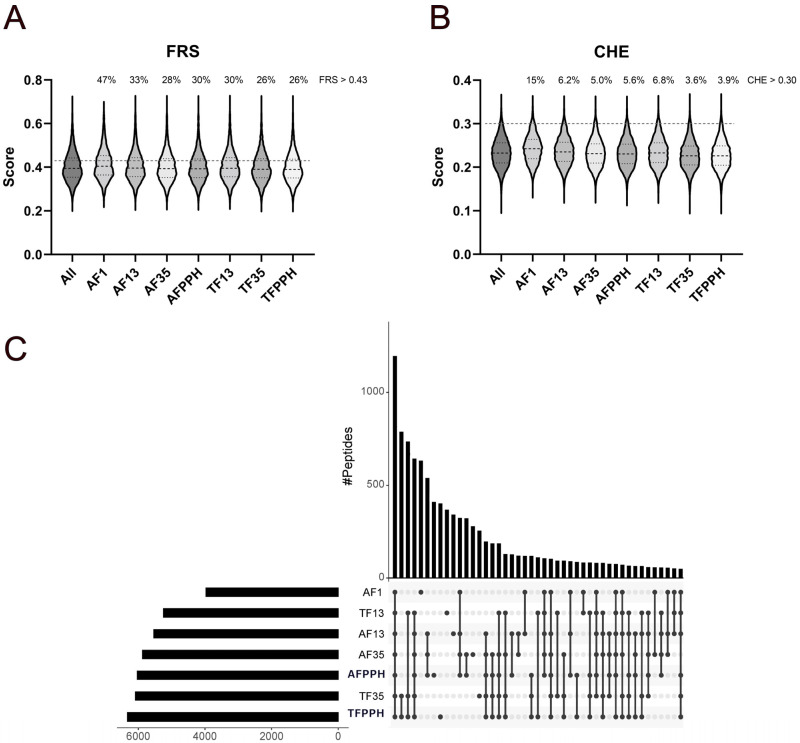
Distribution of FRS (**A**) and CHE (**B**) scores and cross-sample intersects (**C**) for identified peptides in the different PPH fractions. Violin plots show the arithmetic distribution of FRS and CHE scores (**A** and **B**, respectively) with the mean (black dashed line) and upper and lower quartile boundaries (back dotted line) for the individual fractions as well as the threshold scores for the two properties (grey dashed line). The overlap of identified peptides between PPH fractions are illustrated in an UpSet plot (**C**), indicating the arithmetic magnitude of the intersects (i.e., number of shared peptides) as well as the total number of peptide IDs within each fraction. The plot is truncated at an intersect of *n* = 50 for simplicity, but all intersects are shown in the Appendix A.

**Figure 2 antioxidants-13-00346-f002:**
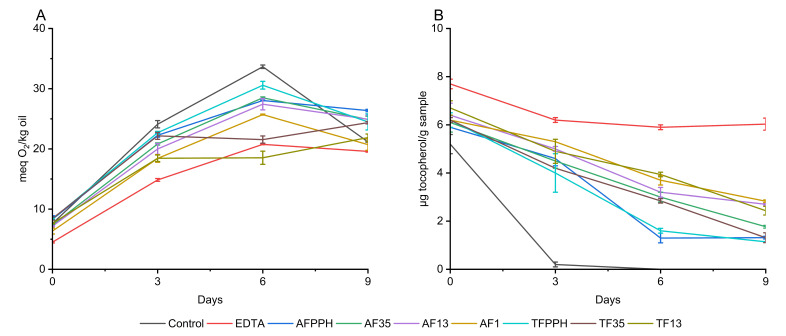
Formation of lipid hydroperoxides (**A**) and consumption of alpha-tocopherol (**B**) during 9 days of storage of 5% fish oil-in-water emulsions.

**Figure 3 antioxidants-13-00346-f003:**
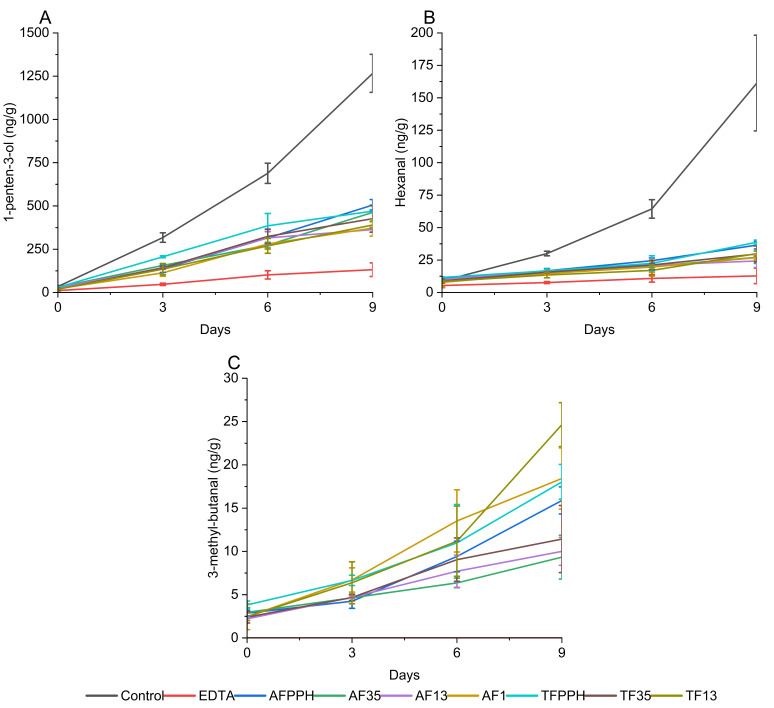
Evolution of secondary volatile oxidation products 1-penten-3-ol (**A**), hexanal (**B**) and 3-methyl-butanal (**C**) in 5% fish oil-in-water emulsions during 9 days of storage.

**Table 1 antioxidants-13-00346-t001:** Enzymatic hydrolysis of potato protein using different enzymes. Values are given as mean ± standard deviation. Different letters (a–c) in the same column indicate a statistically significant difference.

Sample Name	Enzyme(s)	Protein Content of Hydrolysates (%)	Protein Yield (%)	DH (%)
Alc	Alcalase	69 ± 0.01 ^a^	31.9 ± 0.6 ^ab^	20.7 ± 0.7 ^b^
AlcFla	Alcalase and Flavourzyme	88 ± 0.06 ^b^	40.2 ± 4.2 ^b^	38.8 ± 3.6 ^c^
Try	Trypsin	60 ± 0.00 ^a^	27.2 ± 0.6 ^a^	13.1 ± 1.0 ^a^
TryFla	Trypsin and Flavourzyme	86 ± 0.07 ^b^	39.0 ± 4.1 ^b^	38.1 ± 3.0 ^c^

**Table 2 antioxidants-13-00346-t002:** Summary statistics for MS-based peptidomic analysis of fractions. For each fraction, the number of identified peptides (Peptide IDs), arithmetic and intensity-weighted means for peptide length and charge, and relative molar abundance of amino acids based on intensity-weighted peptide-level data are shown. Amino acid composition is color-coded from low (red) to high (green) relative, molar content of individual amino acids.

		AF1	AF13	AF35	AFPPH	TF13	TF35	TFPPH
	Peptide IDs	3968	5535	5872	6032	5241	6085	6325
Length	Mean	6.36	7.75	8.67	8.76	8.17	9.66	9.79
	Weighted mean	4.80	6.57	7.77	7.79	7.91	9.50	9.51
Charge ^1^	Mean	−0.41	−0.68	−0.82	−0.82	−0.72	−1.01	−1.02
	Weighted mean	−0.30	−0.64	−0.83	−0.78	−0.64	−0.98	−1.02
Amino acid composition (relative molar abundance)								
	Ala	4.2%	3.7%	4.0%	3.8%	4.1%	5.0%	4.8%
	Arg	0.7%	1.4%	1.6%	1.5%	1.6%	1.9%	1.6%
	Asn	2.1%	3.5%	3.8%	3.8%	4.2%	4.5%	4.4%
	Asp	4.0%	7.6%	8.4%	7.8%	5.5%	8.0%	7.9%
	Cys	0.2%	0.3%	0.3%	0.6%	0.1%	0.3%	0.4%
	Gln	1.4%	1.9%	2.0%	2.1%	1.3%	1.6%	1.5%
	Glu	2.7%	4.6%	5.8%	5.5%	5.8%	6.6%	6.9%
	Gly	8.2%	9.6%	9.9%	9.7%	11.3%	12.4%	11.2%
	His	0.6%	0.7%	0.6%	0.7%	0.7%	0.6%	0.6%
	Ile	1.3%	3.0%	4.2%	3.8%	4.2%	4.9%	4.9%
	Leu	23.5%	18.3%	15.4%	15.4%	15.1%	11.3%	11.3%
	Lys	0.7%	1.4%	2.1%	2.4%	1.7%	2.5%	2.7%
	Met	0.8%	1.0%	0.9%	0.8%	1.1%	0.9%	1.2%
	Phe	12.8%	7.8%	6.4%	6.4%	6.5%	4.2%	4.6%
	Pro	14.2%	12.4%	11.9%	12.1%	12.9%	10.6%	11.0%
	Ser	2.9%	3.4%	3.8%	4.6%	4.8%	5.7%	5.8%
	Thr	3.3%	4.6%	5.3%	5.4%	5.5%	6.7%	6.4%
	Trp	3.3%	1.6%	1.1%	1.3%	1.1%	0.7%	0.7%
	Tyr	3.6%	3.7%	3.0%	3.3%	2.7%	2.7%	2.8%
	Val	9.4%	9.6%	9.4%	8.8%	9.8%	9.2%	9.3%

^1^ Charge was calculated by individual AA contribution where Asp/Glu contributes −1, Arg/Lys +1, and His +0.1.

**Table 3 antioxidants-13-00346-t003:** Peptide concentration (mM equivalent glycine) determined using the OPA assay, dissociation constant, and standard error of the fit (SE) determined using surface plasmon resonance for the fractionated potato protein hydrolysates.

Sample	Peptide Concentration (mM Eq. Gly for 1 mg/mL PPH)	K_D_ (mM Eq. Gly)	SE (K_D_)
AF1	3.53	0.72	0.43
AF13	2.78	1.41	0.87
AF35	2.41	2.27	0.58
AFPPH	2.41	6.81	2.10
TF13	2.82	4.85	2.40
TF35	2.27	9.19	7.80
TFPPH	2.27	8.16	7.50

## Data Availability

Data are contained within the article and Appendix A.

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
