# Peer review of "Screening for Metal-Chelating Activity in Potato Protein Hydrolysates Using Surface Plasmon Resonance and Peptidomics"

_antioxidants, 2024, doi:10.3390/antiox13030346_

Round 1
Reviewer 1 Report
The aim of this study was to evaluate the potential of SPR as a screening technique for potato protein hydrolysates (PPH) as metal chelating antioxidants. The manuscript is prepared well, and some minor modifications should be considered before revising the manuscript.
1. The English language should be checked.
2. In the introduction, Did you have any potential application in industry? What is the importance of your study?
3. Please re-organize some points in the discussion since some parts were too lengthy.
4. '2.8.3. Determination of secondary, volatile oxidation products by dynamic headspace GC-MS', Please list the GC/MS conditions.
Author Response
The reply is in the attached word document

Reviewer 2 Report
In this manuscript, Bjørlie and co-authors present a comprehensive study regarding the metal chelating activity of potato protein hydrolysates as a potential tool to overcome the metal-catalyzed lipid oxidation, considered to be a major factor in food waste. Using a combination of peptidomics and bioinformatics, the authors predict the presence of metal-chelating and free radical scavenging peptides in the hydrolysate fractions obtained, while also investigating the potential of Surface Plasmon Resonance (SPR) as a screening tool for metal chelating activity in protein hydrolysates.
The manuscript is very well written, bringing forward new and sound data, well supported by main and supplementary materials. There are certain issues that the authors must address before the manuscript can be considered for publication.
- The authors do not differentiate between the metal-chelating traits and the intrinsic antioxidant traits of the peptides. While the metal-chelating properties of peptides are generally acceptable and easy to predict, the antioxidant/radical scavenging properties are mainly restricted to peptides that contain cysteine. In this regard, the authors should differentiate between the protective action of the peptide hydrolysates that are owed to their metal-chelating activity, to their intrinsic antioxidant activity, or both. Under such circumstances, the phrase “metal-chelating antioxidants” (line 12, 78, etc.) may be re-considered.
- Line 94: please provide details about KMC Amba.
- Line 137: please provide a reference for OPA assay.
- Line 177: please define MOPS abbreviation.
Author Response
Reply is in the attached word document

Round 2
Reviewer 2 Report
The authors responded to reviewer's concerns.
The authors responded to reviewer's concerns.